# Subgroup Analysis of TLR2, -3, -4 and -8 in Relation to the Severity of Clinical Manifestations of Cervical HPV Infection

**DOI:** 10.3390/ijms25179719

**Published:** 2024-09-08

**Authors:** Alexander Dushkin, Maxim Afanasiev, Stanislav Afanasiev, Tatiana Grishacheva, Elena Biryukova, Irina Dushkina, Alexander Karaulov

**Affiliations:** 1Project Office, Moscow City Hospital 52, 123182 Moscow, Russia; 2Clinical Immunology and Allergology Department, I.M. Sechenov First Moscow State Medical University (Sechenov University), 119048 Moscow, Russia; afanasev_m_s@staff.sechenov.ru (M.A.); rectorat@staff.sechenov.ru (E.B.); karaulov_a_v@staff.sechenov.ru (A.K.); 3G.N. Gabrichevsky Moscow Research Institute for Epidemiology and Microbiology, Federal Service for the Oversight of Consumer Protection and Welfare, 125212 Moscow, Russia; 4Center of Laser Medicine, First Pavlov State Medical University of St. Petersburg, 192071 Saint-Petersburg, Russia; info@lasmed.spb.ru; 5Gynecology Department, Moscow City Hospital 67, 123423 Moscow, Russia; gkb67@zdrav.mos.ru

**Keywords:** human papillomavirus, Toll-like receptors, squamous intraepithelial lesion, innate immune

## Abstract

We present the findings of assessing the expression levels of extracellular TLR2 and TLR4 and intracellular TLR3 and TLR8 correlating with the severity of clinical manifestations of HPV infection. A total of 199 women took part in a single-center prospective comparative research study on TLR2, TLR3, TLR4 and TLR8 expression in HPV-related cervical lesions. TLRs’ mRNA expression was analyzed using real-time reverse transcription polymerase chain reaction (RT-PCR). Our results indicate the potential significance of TLR3, TLR4 and TLR8 in responding to HPV infection and its progression to SILs and CC, highlighting the importance of HPV polyinfection in relation to TLR4 and TLR8.

## 1. Introduction

Human papillomavirus (HPV) infection remains one of the most prevalent viral infections worldwide with significant implications for public health. HPV infection is identified as the primary cause of infection-related cancers such as cervical, vulvar, vaginal and penile cancers [1]. According to the International Agency for Research on Cancer, a branch of the World Health Organization, cervical cancer (CC) was ranked eighth (662,301 women) among cancers affecting individuals (excluding non-melanoma skin cancer) of both sexes and fourth among cancers specifically affecting women in 2022 [2]. The CC progression stems from the cervical HPV infection, which can present as either asymptomatic viral carriage or clinical manifestations as low-grade (LSIL) and high-grade (HSIL) squamous intraepithelial lesions. These conditions are occurring amidst chronic inflammation. HPV uses human cell polymerases for replication until their differentiation. Virion production necessitates host cell transcription factors expressed during cell differentiation. Consequently, genome replication and virion production must evade immune response to prevent apoptosis. The molecular and genetic mechanisms utilized by HPV to harmonize these conflicting yet synergetic processes may contribute to oncogenesis. According to the theories proposed by Stanley M.A. et al., 2007 and Mirabello L. et al., 2018, cancer emerges as a ‘by-product’ or ‘collateral damage’ of HPV infection [3,4]. Cancer represents the terminal stage for the virus, hindering its capacity for further dissemination by reducing the production of viral particles.

The human body’s protection against pathogens depends on the intricate interplay between the innate and adaptive immune system. Upon pathogen exposure, innate immune cells promptly recognize foreign agents by detecting nonspecific molecular structures such as cell wall lipopolysaccharides, lipoteichoic acids, peptidoglycans, viral RNA and flagellin. This recognition is mediated by pattern recognition receptors [5,6]. In the context of HPV pathogenesis, significant attention has been directed toward its interaction with components of the innate immune response, with particular emphasis on Toll-like receptors (TLRs). TLRs are pivotal in detecting pathogens and triggering the immune response [7]. Humans possess a diverse repertoire of TLRs, each specialized in the recognition of specific pathogens. Among these, TLR3, TLR7, TLR8 and TLR9 are pivotal in the recognition of HPV and the initiation of innate mucosal immune responses [8,9,10]. However, evidence suggests that HPV may influence TLR4 [11,12] and TLR5 [13] in cervical cancer development. TLR3, TLR7, TLR8 and TLR9 are located on the membranes of intracellular organelles, suggesting their interaction with HPV during cellular penetration [14]. However, the specifics of this interaction during the penetration stage remain unclear.

The aim of this study is to present the findings of assessing the expression levels of membrane TLR2 and TLR4 and endosomal TLR3 and TLR8 correlating with the severity of clinical manifestations of HPV infection.

## 2. Results

### 2.1. Group 1 vs. Group 2 vs. Group 3

Upon comparing the expression levels of TLR2, TLR3, TLR4 and TLR8 across Groups 1, 2 and 3, no statistically significant differences were observed (Figure 1).

In a more detailed analysis of each group, we made a notable observation. The expression levels of TLRs showed significant differences in patients with active HPV infection and/or without cervical lesions (Group 2), as presented in Table 1. In the post hoc analysis, TLR2 exhibited significantly higher expression compared to TLR3 (*p* < 0.001) and TLR4 (*p* < 0.001). Additionally, TLR8 showed higher expression compared to TLR3 (*p* < 0.001) and TLR4 (*p* < 0.001). TLR3 expression was notably downregulated compared to TLR4 (*p* = 0.0414). No significant differences were observed between TLR2 and TLR8.

In patients with cervical lesions but without active HPV infection (Group 3) significant differences in TLR expression were observed (*p* = 0.0016). A downregulation of TLR3 was observed in comparison to TLR2 (*p* = 0.0032) and TLR8 (*p* < 0.001). No other TLR expressions exhibited statistically significant differences in the post hoc analysis (*p* > 0.05).

### 2.2. Group 1 vs. Group 2 (HPV^+^/SIL^+^) vs. Group 2 (HPV^+^/SIL^−^) vs. Group 3

A subanalysis of TLR expression was conducted in patients from Group 2, further divided into subgroups: virus carriers (HPV^+^/SIL^−^) and those with HPV-associated cervical lesions (HPV^+^/SIL^+^). They were compared to Groups 1 and 3. No statistically significant changes were observed in the expression of TLR2 (*p* = 0.655), TLR3 (*p* = 0.126), TLR4 (*p* = 0.35) and TLR8 (*p* = 0.697).

We detected a downregulation trend of TLR3 in comparison with Group 1 and Group 3 (*p* = 0.065). The expression of TLR3 was higher by 1.86 times in the subgroup with HPV^+^/SIL^−^ of Group 2 in comparison with Group 3 (*p* = 0.022). No other TLR expressions exhibited statistically significant differences in the pairwise analysis (*p* > 0.05).

### 2.3. Group 1 vs. Group 2 (HPV^+(mono)^/SIL^+^) vs. Group 2 (HPV^+(poly)^/SIL^+^) vs. Group 2 (HPV^+(mono)^/SIL^−^) vs. Group 2 (HPV^+(poly)^/SIL^−^) vs. Group 3

For a more in-depth investigation, a subgroup analysis of Group 2 was conducted based on HPV mono- (HPV^+(mono)^) or polyinfection (HPV^+(poly)^) and cervical lesions (SIL^+^) or the lack thereof (SIL^−^). We did not observe differences in the subgroup with cervical lesions and HPV^+(mono)^ when compared with Group 1 and Group 3 in the expression of TLR2, TLR3, TLR4 and TLR8 (*p* = 0.534, *p* = 0.24, *p* = 0.923 and *p* = 0.469, respectively). The same results were obtained when comparing the subgroup with cervical lesions and HPV^+(poly)^ with Group 1 and Group 3.

In the analysis of TLR expression in the subgroup (HPV^+(mono)^/SIL^+^), a statistically significant hyperexpression of TLR2 and TLR8 was observed, along with the downregulation of TLR3 expression (*p* < 0.001), and is shown in Figure 2.

In the subgroup with HPV^+(mono)^/SIL^−^ of Group 2, significant differences were found in TLR3 expression (*p* = 0.023) compared to Group 1 and Group 3 (Figure 3). Further analysis of TLR expression in subgroups HPV^+(mono)^/SIL^−^ and HPV^+(poly)^/SIL^−^ in comparison with Group 1 and Group 3 demonstrated a downregulation of TLR3 expression (*p* = 0.0482). No differences in TLR expression were observed when cervical lesions were present compared to Group 1 and Group 3 as shown in Table 2.

### 2.4. Group 1 vs. Group 2 (LSIL/HSIL/ICC and HPV^+^) vs. Group 3 (LSIL/HSIL/ICC and HPV^−^)

An analysis of TLR expression in samples was conducted based on the severity of cervical lesions of the HPV infection and healthy women (Group 1) and is shown in Table 3. We divided Group 2 and Group 3 into three subgroups depending on cervical lesion severity. Among the investigated TLRs, suppressed expression of TLR3 was found in patients with HPV^+^/LSIL (*p* = 0.0016) and HPV^+^/HSIL (*p* < 0.001). Hyperexpression of TLR4 was observed in patients with HPV^+^/LSIL and HPV^+^/ICC compared to HPV^+^/HSIL (*p* = 0.0196). When compared to the control group, the expression level of TLR4 was higher in patients with HPV^+^/LSIL and HPV^+^/ICC (*p* = 0.045). The expression level of TLR8 was significantly higher in patients with HPV^+^/ICC compared to HPV^+^/LSIL and HPV^+^/HSIL (*p* = 0.0072) and the control group (*p* = 0.0112).

When comparing TLRs within the LSIL subgroup with HPV^+^ (Figure 4), a significant decrease in TLR3 expression was found compared to TLR4 (*p* = 0.00464) and TLR8 (*p* < 0.001). Additionally, the expression level of TLR8 was higher compared to TLR4 (*p* = 0.0123). Within the HSIL subgroup with HPV^+^, there was a significant decrease in TLR3 expression compared to TLR2, TLR4 and TLR8 (*p* < 0.001, *p* < 0.001 and *p* = 0.0372, respectively). No statistically significant differences in TLR expression were found in patients with clinical manifestations in the absence of active HPV infection.

A pairwise subanalysis of TLR expression was conducted among patients with or without active HPV infection, depending on the severity of clinical manifestations of HPV, and compared with Group 1. Statistically significant differences in TLR8 expression were observed between patients with HPV^+^/LSIL and Group 1 (*p* = 0.049). A trend toward TLR8 hyperexpression was noted in patients with HPV^+^/LSIL compared to the control group (*p* = 0.0804). In ICC associated with active HPV infection, TLR8 hyperexpression was observed to be 2.3 times higher compared to Group 1 (*p* = 0.0205). TLR4 expression was 1.4 times higher in patients with HPV^+^/LSIL compared to HPV^+^/HSIL (*p* = 0.0107). TLR8 hyperexpression was observed in patients with HPV^+^/LSIL compared to HPV^+^/HSIL (*p* = 0.0286). Additionally, a trend toward TLR2 hyperexpression was noted in patients with HPV^+^/LSIL compared to HPV^+^/HSIL (*p* = 0.0546). The expression level of TLR8 was 2.13 times higher in patients with ICC associated with active HPV infection compared to HPV^+^/HSIL (*p* = 0.0109). TLR8 expression was also significantly higher in HPV^+^/ICC compared to HPV^+^/HSIL by 2.12 times (*p* = 0.0109). A trend toward TLR8 hyperexpression was observed when comparing patients with invasive cervical squamous cell carcinoma associated with active HPV infection and HPV^−^/LSIL (*p* = 0.0571).

Furthermore, we conducted pairwise comparisons of TLR expression among samples categorized by mono- and polyinfection of HPV and clinical manifestations (Table 4). Samples obtained from patients with ICC were excluded from this subanalysis due to the limited sample size. Our analysis revealed no statistically significant differences in the expression of TLR2, TLR3, TLR4 and TLR8 in samples with HPV^+(mono)^/LSIL and HPV^+(poly)^/LSIL. Similarly, the expression levels of TLR2, TLR3, TLR4 and TLR8 in samples with HPV^+(mono)^/HSIL and HPV^+(poly)^/HSIL were comparable. However, when comparing the expression of TLR2 between HPV^+(mono)^/LSIL and HPV^+(mono)^/HSIL, we observed a trend toward hyperexpression in HPV^+(mono)^/LSIL (*p* = 0.0619). Additionally, a reduction in the expression of TLR4 and TLR8 was noted in samples from HPV^+(poly)^/HSIL compared to HPV^+(poly)^/LSIL (*p* = 0.0362 and *p* = 0.0318, respectively).

## 3. Discussion

There are still gaps in our understanding of HPV’s role in carcinogenesis. While HPV is widely recognized as the primary cause of invasive cervical cancer (ICC), the question remains: “Is the development of ICC the inevitable outcome of HPV infection?” Typically, a viral infection aims to infect more cells to generate viral particles. However, persistent HPV infection leads to chronic inflammation and precancerous morphological changes [15,16], with the immunomolecular aspect remaining largely unaltered.

HPV is epitheliotropic, infecting the basal layer upon contact with the human epithelium, often through microfissures or areas with actively dividing cells. Epithelial cells serve as the first line of defense against HPV exposure. Given the cervical epithelium’s constant interaction with pathogens, it necessitates a universal mechanism for pathogen recognition and a swift response to eliminate them [17]. Cervical epithelial cells express pattern recognition receptors on their surfaces, detecting molecular patterns associated with pathogens such as bacteria, parasites and viruses. The roles of TLR2, TLR3, TLR4 and TLR8 as predictors of latent herpesvirus infection activation during pregnancy have been explored. Savchenko et al. (2021) concluded that TLR8 may predict the transition from latent herpesvirus infection to the lytic phase [18]. Additionally, the impact of herpes simplex viruses 1 and 2 on the expression of TLR2, TLR3, TLR4 and TLR8 in predicting miscarriage onset has been studied. Karaulov et al. (2018) found that TLR8 hyperexpression occurs in urogenital infections combined with herpes simplex virus 1 and 2 infections [19]. In our study, we evaluated the expression of TLR2, TLR3, TLR4 and TLR8 in cervical papillomavirus infection.

A previous study by Hasan et al. (2007) assessed the mRNA expression of TLR9 in patients with ICC associated with HPV16 infection. The study demonstrated that the oncoproteins E6/E7 lead to a decrease in TLR9 expression in human keratinocytes. Additionally, the study indicated that HPV16 E6 and E7 proteins do not alter TLR9 protein stability but may affect its mRNA levels, resulting in reduced functionality. It was further shown that HPV16 evades the innate immune system by downregulating TLR9 expression, thereby limiting the host’s ability to mount an effective immune response against the virus. This evasion strategy may contribute to HPV16 persistence in the host and the progression of cervical cancer associated with this HPV type [8].

In contrast to its negative impact on the TLR9 system, HPV16 infection was associated with the activation of TLR3, TLR5 and TLR8 pathways in cervical cells. In our study, the evaluation of TLR3 expression revealed that HPV infection was accompanied by higher expression levels compared to cases with the presence of squamous intraepithelial lesions (SILs) but without viral replication. The presence of SILs in the context of active HPV infection was associated with increased expression of TLR2 and TLR8. Additionally, increased TLR4 expression was observed in HPV infections without SILs. The biological significance of this upregulation is not fully understood, but it may represent a compensatory mechanism in infected cells to counteract reduced TLR9 regulation and initiate immune responses [8]. Notably, malignant transformation is accompanied by increased expression of other TLRs, primarily TLR2 and TLR4 [9,12,20]. Our study found that TLR2 expression increases with HPV monoinfection, while TLR3 expression decreases. In cases of HPV polyinfection, TLR8 expression increases. Our study demonstrates that TLR8 exhibits maximum expression in HSIL/HPV- and ICC/HPV+ cases.

In contrast, studies by Kim et al. (2008) and Lee et al. (2007) describe a different scenario. They observed increased mRNA expression of TLR5 and TLR9 in epithelial cells with severe SILs and cervical cancer, while in healthy cells or those with mild SIL, the mRNA expression of TLR5 and TLR9 was minimal or absent [13,21]. Our study complements the understanding of TLR expression in ICC. For instance, TLR8 shows higher expression in ICC compared to LSIL and HSIL. Zidi et al. (2016) investigated the mRNA expression of TLR2, TLR3, TLR4 and TLR9 at different stages (I–IV) of ICC [9]. They found that maximal TLR expression occurs at stages III–IV compared to stages I–II. Our research contributes to understanding HPV infection dynamics, focusing on earlier clinical manifestations, particularly changes in the expression of TLR2, TLR3 and TLR4. Yu et al. describe a decrease in the mRNA expression of TLR4 with increasing severity of SIL [11]. Our findings align with these observations, particularly in HPV-negative cases. The decrease in TLR4 mRNA expression is associated with HPV DNA integration into the host cell DNA, as observed with hyperexpression of p16^INK4a^ in HPV infection with LSIL [11].

## 4. Materials and Methods

### 4.1. Study Cohort

A total of 199 women took part in a single-center prospective comparative research study of TLR2, TLR3, TLR4 and TLR8 expression in HPV-related cervical lesions. The study was performed at the I.M. Sechenov Moscow State Medical University. This study is part of the research project “PDT-induced innate immune response study of patients who were diagnosed with HPV-related cervical diseases and treated by photodynamic therapy”.

The investigation was approved by the local ethical committee of I.M. Sechenov First Moscow State Medical University (Protocol N°15-21/01.09.2021).

#### 4.1.1. Inclusion Criteria

Range of age: 18–65 years;HPV-testing (PCR or HC) of 16, 18, 31, 33, 35, 39, 45, 51, 52, 56, 58, 59, 66 and 68 types;Successful results from the expression analysis of one or both TLRs using OT real-time PCR;Successful liquid-based/traditional cytology performed;Histology investigation for all HSIL species;Signed informed consent to take part in the investigation.

#### 4.1.2. Non-Inclusion Criteria

Out of age range: 18–65 years;Not enough HPV types in HPV-testing panel (16, 18, 31, 33, 35, 39, 45, 51, 52, 56, 58, 59, 66 and 68 types).

#### 4.1.3. Exclusion Criteria

Unsuccessful results from OT real-time PCR TLRs.Absence of signed informed consent to take part in the investigation.

#### 4.1.4. Study Design

The subgroup comparative research study design is shown in Figure 5. All patients were divided into three main groups:Group 1 consists of individuals without HPV infection (HPV^−^), currently undergoing HPV testing and showing negative for intraepithelial lesions or malignancy (NILM) based on cytology investigation;Group 2 comprises individuals with current HPV infection (HPV^+^) undergoing HPV testing regardless of whether intraepithelial lesions or malignancy is detected or not through (SIL^+/−^) cytology investigation;Group 3 consists of individuals without HPV infection (HPV^−^), currently undergoing HPV testing and showing squamous intraepithelial lesions or malignancy (SIL^+^) based on cytology/histology investigation.

### 4.2. Smear Collection

A total of 199 samples of cervical epithelial cells were collected and delivered in the molecular immunology laboratory of FSBSI “I. Mechnikov Research Institute of Vaccine and Sera”. There was an assessment of TLR expression levels based on the OT real-time PCR method. All patients refrained from using local antiseptics and antibacterial and antifungal drugs for two weeks prior to the collection of biological samples. Samples were collected either before menstruation or within two days following its cessation. Patients abstained from cleansing the external genitalia for three hours preceding the study.

### 4.3. Reagents and Instruments of TLR Expression Detection

TLR2, TLR3, TLR4 and TLR8 mRNA expression was analyzed using real-time reverse transcription polymerase chain reaction (RT-PCR). The DT-96 Thermal Cycler (DNA-Technology LLC, Moscow, Russia) was employed for real-time PCR. The expression levels of TLR mRNAs were normalized to the expression level of the GAPDH mRNA. Samples were centrifuged at 10,000 rotations per minute, separating the sediment from the supernatant. The sediment was dissolved using the high-quality solution ‘RIBO-sorb K2-1-Et-100-CE’ (AmpliSense Biotechnology, Moscow, Russia) for RNA extraction from clinical materials. Oligonucleotide primer design for TLR2, TLR3, TLR4 and TLR8 was based on bioinformatic analysis using Vector NTI Advance 9.0 PC software (http://www.invitrogen.com/site/us/en/home.html, accessed on 15 September 2021), DNASTAR and BLAST (http://blast.ncbi.nlm.nih.gov/Blast.cgi/, accessed on 15 September 2021). Primer sequences for GAPDH and TLR expression detection are presented in Table 5. Total mRNA was extracted following the protocol for DNA/RNA extraction using silica gel affinity chromatography from clinical material (ILS LLC, St. Petersburg, Russia). The first-strand cDNA synthesis was performed using the OT-1 kit protocol (Syntol LLC, Moscow, Russia). The cDNA levels in samples were quantified using the cycle threshold (Ct) method with SYBR Green I (Syntol LLC, Moscow, Russia) in real-time PCR.

### 4.4. Calculation of mRNA Expression Level

The TLR2, TLR3, TLR4 and TLR8 mRNA expression of TLRs was analyzed twice. The Ct values provided from OT real-time PCR instrumentation were calculated for average values. The delta-delta Ct method, also known as the 2^−∆∆Ct^, was used to estimate relative fold gene expression. Sixteen samples of cervical epithelial cells (Group 1) were collected for use as calibrators.

### 4.5. HPV Testing

Active HPV infection in patients was identified using PCR to determine the HPV type and quantify its absolute viral load (*n* = 170; 87.46%). HPV testing was conducted during the initial consultation with the local physician in 17 patients (8.54%) using PCR for HPV typing without assessing the absolute viral load. Eight patients (4%) underwent HPV testing using the digene hybrid capture 2 high-risk HPV DNA tets (, QIAGEN, Hilden, Germany). HPV testing of cervical specimens was performed in specialized clinical diagnostic laboratories, which ensured research quality through adherence to a certified quality management system in compliance with international standards [22,23].

### 4.6. Morphology Investigations

The presence and degree of intraepithelial lesions were determined using liquid cytology, specifically the BD SurePath liquid-based Pap test (BD SurePath, Franklin Lakes, NJ, USA, SKU/REF 491452) with the BD FocalPoint GS automated viewing system (BD FocalPoint GS Imaging System, Franklin Lakes, NJ, USA, PID No. 779-04194-02), or through traditional cytological examination. If results were suspicious for invasive cancer or showed cellular composition indicative of carcinoma in situ, patients underwent an incisional biopsy using the MAGNUM Automatic Reusable Biopsy System (Bard Magnum, Tempe, AZ, USA, RZN 2015/2501), followed by histological examination to determine the extent of the lesion.

### 4.7. Statistical Analysis

Data collection and storage were managed using Microsoft Office 365 software package Excel v.16.88. We utilized Python 3.12 statistical packages within the IDE Visual Studio Code 1.76.1 (Universal) for math and statistical data processing.

We assessed quantitative variables for adherence to normal and non-normal distributions using the Shapiro–Wilk test. All quantitative variables had a non-normal distribution. Descriptive statistics such as median (Me) and interquartile interval (IQR) were utilized for quantitative variables demonstrating a non-normal distribution. Categorical variables were described using absolute values (*n*) and percentages (%). A comparison of two independent groups based on quantitative indicators was conducted using the Mann–Whitney U test. For the comparison of three or more independent groups, the Kruskal–Wallis H test was employed followed by post hoc analysis using Dunn’s test with Bonferroni *p* adjustment. For two dependent groups, the Wilcoxon signed-rank test was utilized, while the Friedman *χ*^2^ test was applied for comparisons involving three or more dependent groups. Post hoc analysis was conducted by using the Wilcoxon signed-rank test. A comparison of proportions in the analysis of fourfold tables and contingency tables of categorical variables was carried out using Pearson’s *χ*^2^ test. Table 6 presents an analysis of the clinical and anamnestic characteristics of the patients included in the study.

## 5. Conclusions

This paper presents findings from evaluating TLR2, TLR3, TLR4 and TLR8 expression in patients with cervical HPV infection and its associated cervical lesions (incl. cervical cancer). We assessed TLR expression in both HPV mono- and polyinfection. Our results indicate the potential significance of TLR3 and TLR8 (endosomal receptors) and TLR4 (membrane receptor) in responding to HPV infection and its progression to SILs and ICC, highlighting the importance of HPV polyinfection in relation to TLR4 and TLR8.

However, this study has certain limitations. The sample size was relatively small, which may limit the generalizability of our findings. Additionally, the study’s cross-sectional design does not allow for establishing causal relationships between TLR expression and HPV progression. Further longitudinal studies with larger cohorts are necessary to confirm these results and to explore the underlying mechanisms of TLR involvement, particularly at the protein level and their associated cytokine responses, in HPV-related cervical pathology. We also did not assess the corresponding TLR proteins and related cytokines, which are critical in mediating the immune response to HPV infection. The patterns of cytokine expression associated with different TLRs suggest a complex interplay between these receptors and the host immune system, potentially influencing the outcome of HPV infection and its progression to cervical lesions.

A comprehension of the interplay between TLRs and HPV infection plays a crucial role in infection management and effective control. Investigation in this domain aids in uncovering novel immune control mechanisms for HPV infection and contributes to the advancement of innovative approaches for the diagnosis, treatment and prevention of this pervasive viral infection.

## Figures and Tables

**Figure 1 ijms-25-09719-f001:**
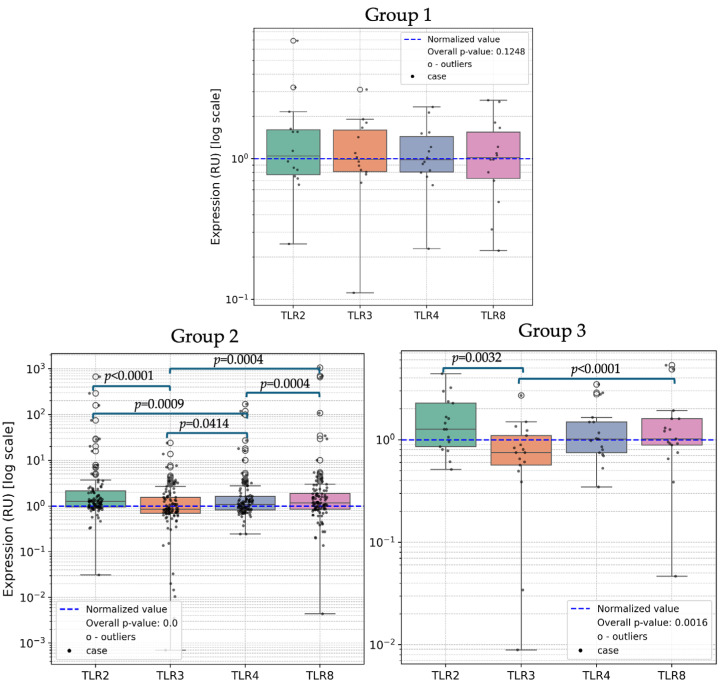
Comparison of TLRs in Group 1, Group 2 and Group 3. Note: RU—relative units; TLR—Toll-like receptor.

**Figure 2 ijms-25-09719-f002:**
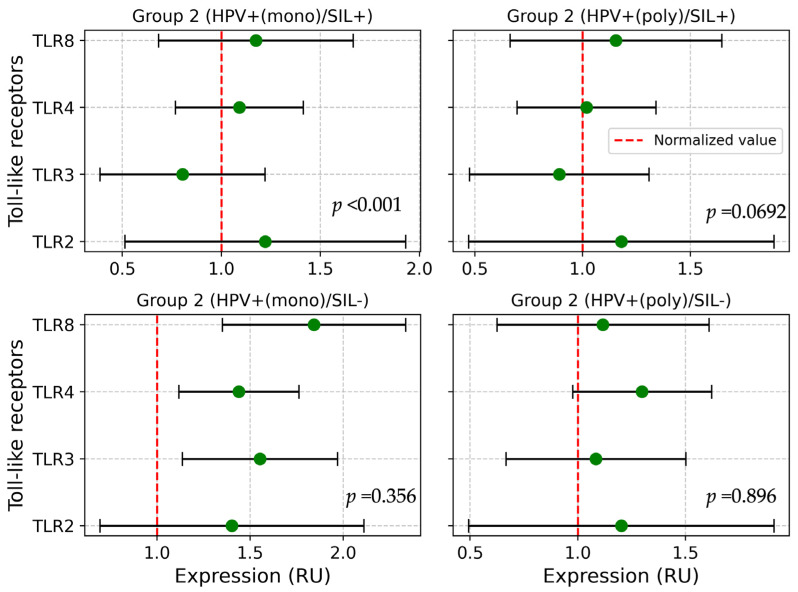
Comparison of TLRs in subgroups of Group 2. Note: HPV—human papillomavirus; SIL—squamous intraepithelial lesion; RU—relative units; TLR—Toll-like receptor.

**Figure 3 ijms-25-09719-f003:**
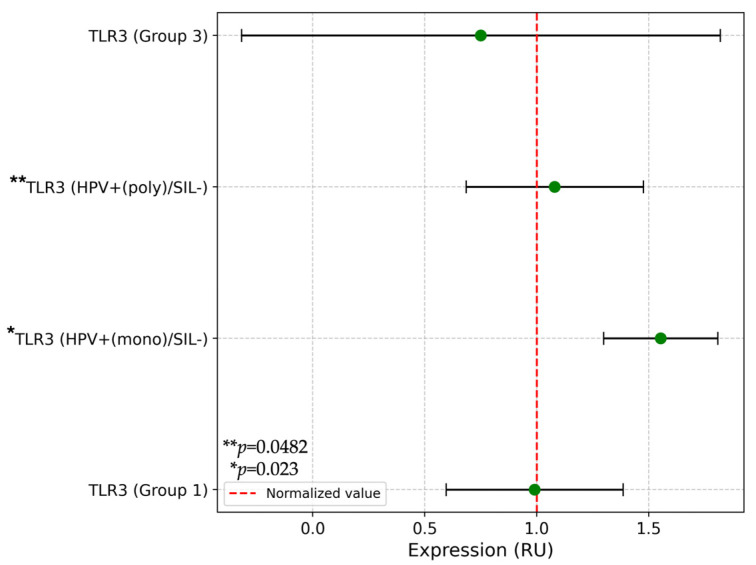
Comparison of TLR3 in subgroups (HPV^+(mono)^/SIL^−^ and HPV^+(poly)^/SIL^−^) of Group 2. Note: HPV—human papillomavirus; SIL—squamous intraepithelial lesion; RU—relative units; TLR—Toll-like receptor. * Group 1 vs. subgroup HPV^+(mono)^/SIL^−^ vs. Group 3. ** Group 1 vs. subgroup HPV^+(mono)^/SIL^−^ vs. subgroup HPV^+(poly)^/SIL vs. Group 3.

**Figure 4 ijms-25-09719-f004:**
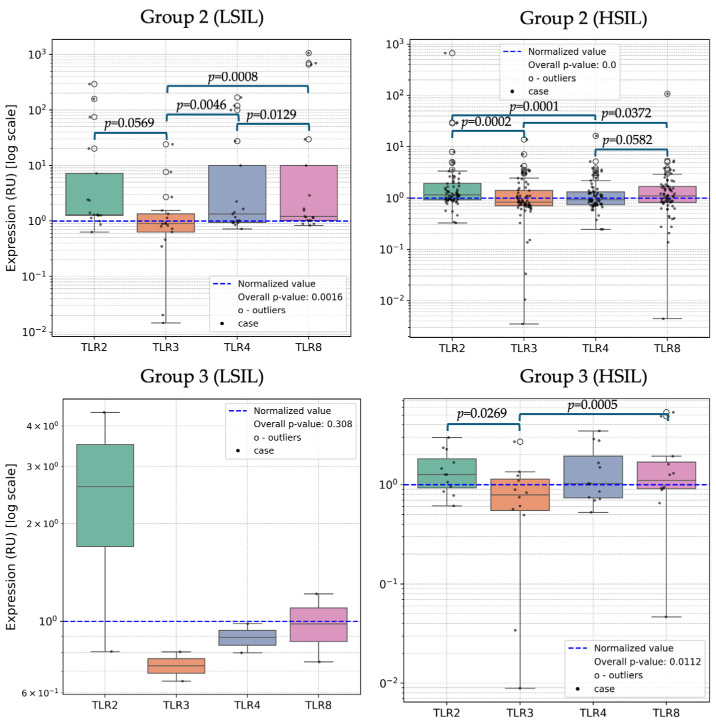
Comparison of TLRs in Group 2 (LSIL) and Group 2 (HSIL). Note: HPV—human papillomavirus; HSIL—high-grade squamous intraepithelial lesion; LSIL—low-grade squamous intraepithelial lesion; RU—relative units; TLR—Toll-like receptor.

**Figure 5 ijms-25-09719-f005:**
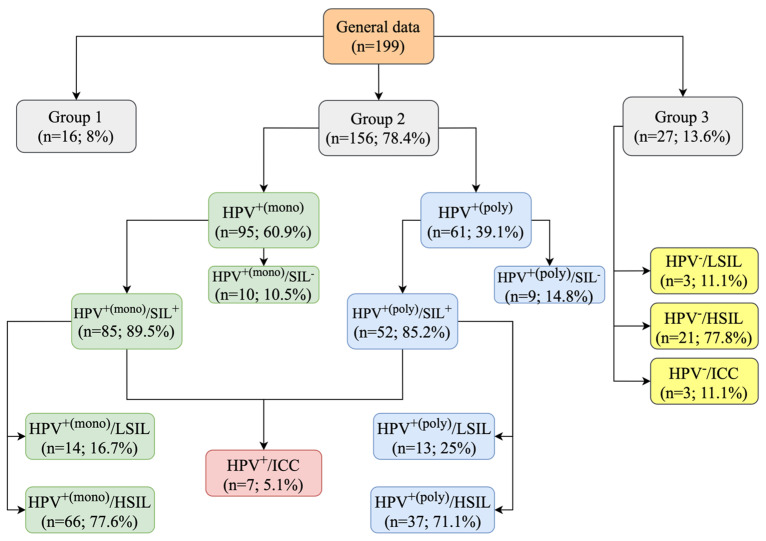
Study design; HPV—human papillomavirus; HPV^+(mono)^—one HPV was detected; HPV^+(poly)^—more than one HPV was detected; HPV^−^—no HPV detected; HSIL—high-grade squamous intraepithelial lesion; ICC—invasive cervical cancer; LSIL—low-grade squamous intraepithelial lesion.

**Table 1 ijms-25-09719-t001:** TLR expression analysis between groups.

TLRs	Group 1	Group 2	Group 3	*p* Value
TLR2 (RU)	1.05 [0.8; 1.6]	1.26 [0.96; 2.2]	1.26 [0.85; 2.28]	0.476
TLR3 (RU)	0.99 [0.8; 1.6]	0.86 [0.68; 1.55]	0.75 [0.57; 1.1]	0.201
TLR4 (RU)	0.98 [0.8; 1.4]	1.09 [0.83; 1.54]	1.49 [0.76; 1.49]	0.663
TLR8 (RU)	1.02 [0.72; 1.55]	1.17 [0.86; 1.84]	1.12 [0.9; 1.53]	0.505
*p* value	0.1248	<0.001	0.0016	

RU—relative units.

**Table 2 ijms-25-09719-t002:** Comparing TLR expression in HPV^+(mono)^ and HPV^+(poly)^ subgroups of Group 2.

	TLR2 (RU)	TLR3 (RU)	TLR4 (RU)	TLR8 (RU)	*p* Value *
Group 2:					
HPV^+mono^	1.26 [0.95; 2.36]	0.8 [0.68; 1.55]	1.15 [0.77; 1.56]	1.2 [0.86; 2.1]	<0.001
SIL^+^	1.22 [0.94; 2.36]	0.8 [0.64; 1.47]	1.09 [0.75; 1.39]	1.17 [0.86; 1.84]	<0.001
SIL^−^	1.4 [1.31; 1.92]	1.55 [1.4; 1.91]	1.44 [1.25; 1.71]	1.84 [0.95; 2.35]	0.356
HPV^+poly^:	1.18 [0.96; 1.9]	0.89 [0.72; 1.43]	1.05 [0.86; 1.49]	1.15 [0.83; 1.62]	0.089
SIL^+^	1.18 [0.94; 1.95]	0.89 [0.72; 1.29]	1.02 [0.86; 1.57]	1.15 [0.85; 1.83]	0.0692
SIL^−^	1.2 [1; 1.44]	1.08 [0.78; 1.57]	1.3 [1.13; 1.42]	1.34 [1.12; 1.6]	0.896
** *p* value	0.894	0.851	0.249	0.483	

HPV—human papillomavirus; RU—relative units; TLR—Toll-like receptor. * Significant *p*-values for Group 2, stratified by presence or absence of cervical lesion, compared to Group 1. ** Significant *p*-values for subgroups (HPV^+(mono)^ and HPV^+(poly)^) of Group 2 comparison with Group 1 and Group 3.

**Table 3 ijms-25-09719-t003:** TLR expression analysis based on severity of cervical lesions with HPV^+^ or HPV^−^.

	TLR2 (RU)	TLR3 (RU)	TLR4 (RU)	TLR8 (RU)	*p* Value *
L-SIL: HPV^+^	1.31 [1.26; 7.15]	0.89 [0.63; 1.35]	1.34 [0.95; 10.02]	1.22 [0.95; 10.07]	0.0016
HPV^−^	2.6 [1.7; 3.5]	0.73 [0.69; 0.77]	0.89 [0.85; 0.94]	0.98 [0.88; 1.1]	0.308
H-SIL: HPV^+^	1.16 [0.93; 1.97]	0.83 [0.68; 1.37]	0.94 [0.75; 1.32]	1.1 [0.82; 1.63]	<0.001
HPV^−^	1.26 [0.93; 1.82]	0.79 [0.55; 1.13]	1.02 [0.75; 1.65]	1.22 [0.92; 1.6]	0.0112
ICC: HPV^+^	1 [0.94; 3.5]	1.3 [0.39; 2.78]	1.34 [0.91; 4.13]	2.34 [1.5; 4.89]	0.0738
HPV^−^	1.6 [1.06; 2.4]	0.75 [0.57; 1.12]	1.17 [0.76; 1.33]	1.02 [0.7; 1.3]	0.1218
^1^ *p* value *	0.179	0.776	0.045	0.0112	
^2^ *p* value *	0.912	0.3014	0.923	0.8921	

HPV—human papillomavirus; HSIL—high-grade squamous intraepithelial lesion; ICC—invasive cervical cancer; IQR—interquartile range; LSIL—low-grade squamous intraepithelial lesion; Me—median; TLR—Toll-like receptor. Note: ^1^ Group 1 vs. LSIL and HPV^+^ vs. HSIL and HPV^+^ vs. ICC and HPV^+^; ^2^ Group 1 vs. LSIL and HPV^+^ vs. HSIL and HPV^+^ vs. ICC and HPV^+^. * Significant *p*-values for Group 2, stratified by severity, compared to Group 1.

**Table 4 ijms-25-09719-t004:** TLR expression analysis based on HPV-associated cervical lesion severity and HPV^+(mono)^/HPV^+((poly)^ subgroups of Group 2.

TLRs	HPV^+^/LSIL		HPV^+^/HSIL		*p* Value
	HPV^+(mono)^	HPV^+(poly)^	HPV^+(mono)^	HPV^+(poly)^	
TLR2	1.88 [1.29; 2.42]	1.31 [1.06; 13.69]	1.12 [0.93; 2.02]	1.18 [0.94; 1.87]	^1^ 0.0619
					^2^ 0.425
					^3^ 0.512
					^4^ 0.679
TLR3	0.97 [0.46; 1.45]	0.89 [0.68; 1.14]	0.8 [0.68; 1.4]	0.89 [0.72; 1.1]	^1^ 0.918
					^2^ 0.796
					^3^ 0.96
					^4^ 0.351
TLR4	1.26 [1.05; 2.03]	1.44 [0.93; 18.7]	0.93 [0.73; 1.37]	0.98 [0.77; 1.24]	^1^ 0.197
					^2^ 0.0362
					^3^ 0.92
					^4^ 0.828
TLR8	1.19 [1.06; 2.47]	1.55 [1.08; 19.78]	1.11 [0.86; 1.7]	1.02 [0.63; 1.45]	^1^ 0.398
					^2^ 0.0318
					^3^ 0.725
					^4^ 0.344

HPV—human papillomavirus; HSIL—high-grade squamous intraepithelial lesion; LSIL—low-grade squamous intraepithelial lesion; TLR—Toll-like receptor. Note: ^1^ HPV^+(mono)^/LSIL vs. HPV^+(mono)^/HSIL; ^2^ HPV^+(poly)^/LSIL vs. HPV^+(poly)^/HSIL; ^3^ HPV^+(mono)^/LSIL vs. HPV^+(poly)^/LSIL; ^4^ HPV^+(mono)^/HSIL vs. HPV^+(poly)^/HSIL.

**Table 5 ijms-25-09719-t005:** Primer sequences of GAPDH and TLRs.

Genes		Sequence 5′-3′
*GAPDH*	F (5′sense)	TGC-MTC-CTG-CAC-CAC-CAA-CT
	R (3′antisence)	YGC-CTG-CTT-CAC-CAC-CTT-C
*TLR2*	F (5′sense)	CTT-CAC-TCA-GGA-GCA-GCA-AGC
	R (3′antisence)	TGG-AAA-CGG-TGG-CAC-AGG-AC
*TLR3*	F (5′sense)	GGT-CCC-AGC-CTT-ACA-GAG-AA
	R (3′antisence)	AGC-CCG-TGC-AAA-GAG-TGA-GA
*TLR4*	F (5′sense)	GAA-GGG-GTG-CCT-CCA-TTT-CAG-C
	R (3′antisence)	TGC-CTG-AGC-AGG-GTC-TTC-TCC-A
*TLR8*	F (5′sense)	TAG-TTT-CAG-TGG-CAA-TCG-C
	R (3′antisence)	GAG-ACG-AGG-AAA-CTG-CTG-GA

GAPDH—glyceraldehyde-3-phosphate dehydrogenase (house-keeping gene); TLR—Toll-like receptor; F—forward; R—reverse.

**Table 6 ijms-25-09719-t006:** Clinical characteristics of the papillomavirus infection patients and Group 1.

Variable	General Groups			*p* Value
	Group 1	Group 2	Group 3	
Age	35 [30; 40]	33 [28; 39]	37 [30.5; 39]	0.182
Weight	60 [57; 75]	60 [55; 63]	60 [52; 70]	0.660
Menses	14 [12; 14]	13 [12; 14]	13 [13; 14]	0.709
Sexual life onset	18 [17.5; 19.5]	18 [17; 19]	18 [17; 19]	0.381
Number of sexual partners	3 [2.5; 4.5]	5 [3; 7]	5 [3; 9]	0.299
Number of pregnancies	1 [0; 1]	1 [0; 2]	2 [1.25; 3]	0.006
				** 0.012
				*** 0.011
Number of childbirths	0 [0; 1]	1 [0; 1]	1 [1; 2]	0.003
				** 0.019
				*** 0.004
Number of abortions	0 [0; 0]	0 [0; 1]	0.5 [0; 1]	0.158
Number of HPV types	N/A	1 [1; 2]	N/A	N/A
Complaints	2 (13.3%)	27 (17.3%)	3 (11.1%)	0.688
Therapeutical intervention	8 (53.3%)	63 (40.4%)	7 (25.9%)	0.189
Surgical intervention	5 (33.3%)	39 (25%)	13 (48.1%)	0.04
				*** 0.041
Gyn. diseases	13 (86.7%)	116 (74.4%)	21 (77.8%)	0.549
Gyn. infection	3 (20%)	20 (12.8%)	7 (25.9%)	0.185
Previous PVI (>6 months)	N/A	92 (59%)	14 (51.9%)	*** 1
AWE	4 (25%)	121 (80,1%)	18 (66.7%)	
AWE + m + p	1 (6.2%)	69 (45.7%)	11 (40.7%)	0.010
				* 0.007
				** 0.030
TZ:				0.863
Type 1	4 (25%)	36 (23.4%)	6 (22.2%)	
Type 2	4 (25%)	58 (37.7%)	9 (33.3%)	
Type 3	8 (50%)	60 (39%)	12 (44.4%)	

AWE—acetowhite epithelium, Gyn.—gynecological, HPV—human papillomavirus infection, TZ—transformation zone, PVI—papillomavirus infection. Note: * Post-hoc differences between Group 1 and Group 2; ** Post-hoc differences between Group 1 and Group 3; *** Post-hoc differences between Group 2 and Group 3.

## Data Availability

The data and material used and analyzed in the study are available from the corresponding authors upon reasonable request.

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
