# Peer review of "Subgroup Analysis of TLR2, -3, -4 and -8 in Relation to the Severity of Clinical Manifestations of Cervical HPV Infection"

_ijms, 2024, doi:10.3390/ijms25179719_

Round 1

Reviewer 1 Report

Comments and Suggestions for Authors

In their manuscript, Dushkin et al. report a single-center prospective study investigating the correlation between TLR2, TLR3, TLR4 and TLR8 expression and severity of HPV-related cervical lesions. The authors highlight the role of deregulation of TLRs expression in HPV oncogenesis and Its correlation with the HPV induced lesions. The manuscript presents original and good quality scientific data. The form of the manuscript is well written.  The data reported in the manuscript are original and represent a scientific contribution to the field.

The subject of the work is important; however, before it is considered for publication there are several points of the manuscript that needs to be address to in the following aspects:

Major comments

-          The selection of the study population should be clarified since it may influence the results of the study

-          The inclusion and exclusion criteria are unclear – do they refer to the study population or to the validity of the study (e.g. Absence of signed informed consent should be exclusion criteria…).

-          The stratification of study population is not clear –which one is the control group (group 1 or group 3); If the HPV negative patients have been separated in 2 groups based on cytology status why It was not done for HPV positive women.

-          The presentation of the results is not clear (Figures 1 and 2); Figures and tables do not follow the text.

Minor comments:

Material and method section:

-          Primer sequences for TLRs expression detection are not presented

Disscusion section:

-          Page 7, Line 230 to 232 – The sentence “ The decrease in mRNA expression…..” - Add literature data

Comments on the Quality of English Language

Editing of English language required (too long sequences, verb tenses incorrect, the use of colloquial instead of scientific terminology...) 

Author Response

Dear Reviewer! Thank you for your time and critical estimation of our paper.

Comment 1: The selection of the study population should be clarified since it may influence the results of the study.

Answer 1: Groups characteristics are presented in Table 6

Comment 2: The inclusion and exclusion criteria are unclear – do they refer to the study population or to the validity of the study (e.g. Absence of signed informed consent should be exclusion criteria…).

Answer 2: Criteria were corrected.

Comment 3: The stratification of study population is not clear –which one is the control group (group 1 or group 3); If the HPV negative patients have been separated in 2 groups based on cytology status why It was not done for HPV positive women.

Answer 3: Sure, we made this separation in the section results paragraph 2.4. Also, addition information about Groups was add in paragraph 4.1.4.

Comment 4: The presentation of the results is not clear (Figures 1 and 2); Figures and tables do not follow the text.

Answer 4: Information to the text were inserted.

Comment 5: Primer sequences for TLRs expression detection are not presented

Answer 5: Information about primers was added.

Comment 6: Page 7, Line 230 to 232 – The sentence “ The decrease in mRNA expression…..” - Add literature data

Answer 6: Literature data was added.

All sentences were rechecked and corrected for English grammar and readability.

Reviewer 2 Report

Comments and Suggestions for Authors

Dear Authors, 

Cervical cancer is the fourth most frequently diagnosed cancer and the fourth most common cause of cancer death in women worldwide that is why emerging treatment tools such as TLR agonist therapy could provide new hope for HPV - carcinogenesis problem.

The article evaluates more  TLRs and indicate the potential significance of TLR3, TLR4 and TLR8 in responding to HPV infection. Although research on TLRs is still in its early stages  this is a current area can provide new perspectives in HPV related neoplasia.

The manuscript is overall well written and the author brings news in an area of great interest. 

As regarding the drafting the text:

Ø  many references are written without following the intructions

Ø  there are some drafting errors ex.  paragraph 43 -  introducing a word in Russian into the text but also misintroducing the authors

I recommend accepting after a minor revision.

Author Response

Dear Reviewer! Thank you for your time to check our paper and make suggestion to improve it.

Comment 1: many references are written without following the intructions

Answer 1: All references were checked and corrected with Journal rules.

Comment 2: there are some drafting errors ex.  paragraph 43 -  introducing a word in Russian into the text but also misintroducing the authors

Answer 2: Text was revised and corrected. 

Reviewer 3 Report

Comments and Suggestions for Authors

In the present manuscript, Dushkin et al. evaluated the expression level of TLR-2, 3, 3, and 8 in relation to the severity of the clinical manifestation of the cervical HPV infection. Although the study seems to be important, it has a lot of mistakes. The study aims to assess the expression levels of extracellular and intracellular TLRs. However, assessment of the expression level based only on the real-time PCR data can not achieve the aims. Real-time PCR cannot differentiate extracellular or intracellular expression. The authors should perform protein-level expression analyses. Flow cytometry analysis can be applied. 

The manuscript has a lot of mistakes in presenting the result. A thorough revision should be done. Some figures/tables are not mentioned anywhere in the text. In some result sections, results are written very well but no figures or tables are mentioned. Some tables like 1, and 6, do not reflect what is in the text. Presentation of the significant data in graphs would be considered for easy understanding. Results are hard to understand due to many numbers and superscripts in the tables and a lack of clarity in the text. Overall, the manuscript needs a rigorous revision.

Author Response

Dear Reviewer! Thank for your time and review our paper. You comment help us to make better this research.

Comment 1: Real-time PCR cannot differentiate extracellular or intracellular expression. The authors should perform protein-level expression analyses. Flow cytometry analysis can be applied. 

Answer 1. You are right. We made a mistake in the phrases aim. We were corrected in the "... findings of assessing the expression levels of membrane TLR2, TLR4 and endosomal TLR3, TLR8 ..."

Comment 2: The manuscript has a lot of mistakes in presenting the result. A thorough revision should be done. Some figures/tables are not mentioned anywhere in the text. In some result sections, results are written very well but no figures or tables are mentioned. Some tables like 1, and 6, do not reflect what is in the text. Presentation of the significant data in graphs would be considered for easy understanding. Results are hard to understand due to many numbers and superscripts in the tables and a lack of clarity in the text. Overall, the manuscript needs a rigorous revision.

Answer 2: We revised our article. Add figures and new information in tables. Also, we corrected the No. of tables and Figures.

Round 2

Reviewer 3 Report

Comments and Suggestions for Authors

The authors have improved the manuscript. However, it still needs major modification. 

- Remove the reverse "N" symbol after "2007" on line 43.

-Change "TKR2" to TLR2 on line 75

-Lines 71-84 have mentioned many significant P-values. I believe those comparisons are in figure one but not properly highlighted. Those can be highlighted in the form of an asterisk or value on the corresponding figure. That will make it easier to understand.

-The figures in figure 1 are the same as those in the figure 2. Figure 1 should focus on the comparison in each group. Figure 2 should represent the comparison between the groups as in the text. Otherwise, Both the figures look almost the same and redundant. Significance should be highlighted in the figure. 

-What do those circular points in the figures represent? It should be mentioned in the figure legend.

-Using the preposition "to" on line 97 is not correct. 

-Lines 108-122 need to be rephrased. Some sentences are incomplete and some are hard to understand.

- Use consistent terminology in the figure and legend. For example, in Figure 2, the term "control" is used in the figure while "group 1" is used in the legend. Check all the figures to make sure the terminology in the text, figures, and legends are consistent.
- on lines 131-132, TLR expression comparison between the healthy women and HPV is mentioned to be highlighted in Table 3. However, no values for group 1 are included in the table. Or, is the title of the table not clear?

-Include significant p-values in the table 3. Many significant values are mentioned in the text but corresponding values are not shown in the table. Include the p-values for easy understanding as in Table 4.

- The abbreviation "PVI" popped up in the Table 3 title. But it was mentioned nowhere in the text. The use of consistent terminology or abbreviations should be considered very critical. 

-I feel the use of cervical smear as a sample for the analysis of TLR expression using real-time PCR is not ideal. How do you verify this? Do you have any earlier publications to support this? The use of cells or tissues for the isolation of DNA/RNA is ideal for the study. Earlier publications (some of them are cited in this manuscript) used either cells or tissue for PCR analysis or Immunohistochemistry for a similar kind of study. Correct sampling is very important for reliable data.

-References 13 and 21 are the same.

Comments on the Quality of English Language

-Some sentences are incomplete. 

-Some texts need to be rephrased (as pointed out in the comment).

-incorrect use of prepositions was found.

-proper punctuation is needed in some sentences to make meaningful sentences. 

Author Response

Dear reviewer! Thank you for your time and comments that helped us to improve our paper. Answers to your comments below.

Comment 1: - Remove the reverse "N" symbol after "2007" on line 43.

Answer 1: Corrected.

Comment 2: -Change "TKR2" to TLR2 on line 75

Answer 2: Changed.

Comment 3: -Lines 71-84 have mentioned many significant P-values. I believe those comparisons are in figure one but not properly highlighted. Those can be highlighted in the form of an asterisk or value on the corresponding figure. That will make it easier to understand.

Answer 3: P value highlights added. Also figure 1 was modified.

Comment 4: -The figures in figure 1 are the same as those in the figure 2. Figure 1 should focus on the comparison in each group. Figure 2 should represent the comparison between the groups as in the text. Otherwise, Both the figures look almost the same and redundant. Significance should be highlighted in the figure.

Answer 4: Definitely you are right, we deleted this figure.

Comment 5: -What do those circular points in the figures represent? It should be mentioned in the figure legend.

Answer 5: Circles are outliers for boxplots visualization. Legend added in the figure 1

Comment 6: -Using the preposition "to" on line 97 is not correct.

Answer 6: ‘to compare’ replace on word ‘comparison’

Comment 7: -Lines 108-122 need to be rephrased. Some sentences are incomplete and some are hard to understand.

Answer 7: Sentences were rephrased and corrected

Comment 8: - Use consistent terminology in the figure and legend. For example, in Figure 2, the term "control" is used in the figure while "group 1" is used in the legend. Check all the figures to make sure the terminology in the text, figures, and legends are consistent.

Answer 8: All legends were corrected.

Comment 9: - on lines 131-132, TLR expression comparison between the healthy women and HPV is mentioned to be highlighted in Table 3. However, no values for group 1 are included in the table. Or, is the title of the table not clear?

Answer 9: Sure, Table name is unclear. We added the “*Significant p-values for Group 2, stratified by severity, compared to Group 1.”

Comment 10: -Include significant p-values in the table 3. Many significant values are mentioned in the text but corresponding values are not shown in the table. Include the p-values for easy understanding as in Table 4.

Answer 10: P values were added. Also, Figure for visualization.

Comment 11: - The abbreviation "PVI" popped up in the Table 3 title. But it was mentioned nowhere in the text. The use of consistent terminology or abbreviations should be considered very critical.

Answer 11: Abbreviation was deleted.

Comment 12: - I feel the use of cervical smear as a sample for the analysis of TLR expression using real-time PCR is not ideal. How do you verify this? Do you have any earlier publications to support this? The use of cells or tissues for the isolation of DNA/RNA is ideal for the study. Earlier publications (some of them are cited in this manuscript) used either cells or tissue for PCR analysis or Immunohistochemistry for a similar kind of study. Correct sampling is very important for reliable data.

Answer 12: Sure, we used cervical epithelial cells. We replaced this colocation “cervical smear” to “samples of cervical epithelial cells” – paragraph 4.2 and 4.4.

Comment 13: -References 13 and 21 are the same.

Answer 13: Reference 21 deleted.

Comments on the Quality of English Language

Comment 14: -Some sentences are incomplete. 

Comment 15: -Some texts need to be rephrased (as pointed out in the comment).

Comment 16: -incorrect use of prepositions was found.

Comment 17: -proper punctuation is needed in some sentences to make meaningful sentences. 

Answer 14 - 17: Quality of English Language improved

Round 3

Reviewer 3 Report

Comments and Suggestions for Authors

The manuscript has been revised very well. I would still suggest doing a minor modification, especially section 2.3 (lines 103-122). It is still hard to understand this section. Table 2 does not reflect the text properly or vice versa. We usually include the data that are significant either in the form of a table or figure. However, in this section, texts say something and the table shows something else. I don't see any significant values that are mentioned in the text (for example: p=0.023 on line 111, p=0.0482 on line 118) in the table. The authors should give importance to the significant data and show it either as a figure or table. Other insignificant data can be provided as supplementary. Instead, the table reflects only the insignificant data and has no connection with the text. I see one significant value (<0.001 at the extreme right column against the SIL+ subgroup of HPV+mono). However, It is not highlighted what that significant value represents. It is mentioned in the figure legend with an asterisk but nothing in the text (or may be mentioned but not clear). 

Lines 107-108 say "analysis of TLR expression in both subgroups, a statistically significant ........(Table 2)": I don't see any significance in the comparison between the subgroups in Table 2. Which subgroup has hyperexpression? This should be mentioned in the text. For example, "the expression of TLRs was compared between the two subgroups (A and B) and observed hyperexpression of TLR2 (p-value) and TLR8 (p-value) in group A and downregulation of TLR3 (p-value) in group B (Table 2). The table should reflect all these findings. The table or figure should reflect whatever is in the text. The other insignificant data can be included as supplementary data.

Unlike Table 2, the authors have presented Tables 3 and 4, and are easily understandable as both tables have reflected what is written in the text. 

Author Response

Dear Reviewer! I'm pleased to inform and thanks for you time.

Your comments help us to improve our research paper and make significant contributions.

Comment: 

 would still suggest doing a minor modification, especially section 2.3 (lines 103-122). It is still hard to understand this section. Table 2 does not reflect the text properly or vice versa. We usually include the data that are significant either in the form of a table or figure. However, in this section, texts say something and the table shows something else. I don't see any significant values that are mentioned in the text (for example: p=0.023 on line 111, p=0.0482 on line 118) in the table. The authors should give importance to the significant data and show it either as a figure or table. Other insignificant data can be provided as supplementary. Instead, the table reflects only the insignificant data and has no connection with the text. I see one significant value (<0.001 at the extreme right column against the SIL+ subgroup of HPV+mono). However, It is not highlighted what that significant value represents. It is mentioned in the figure legend with an asterisk but nothing in the text (or may be mentioned but not clear). 

Lines 107-108 say "analysis of TLR expression in both subgroups, a statistically significant ........(Table 2)": I don't see any significance in the comparison between the subgroups in Table 2. Which subgroup has hyperexpression? This should be mentioned in the text. For example, "the expression of TLRs was compared between the two subgroups (A and B) and observed hyperexpression of TLR2 (p-value) and TLR8 (p-value) in group A and downregulation of TLR3 (p-value) in group B (Table 2). The table should reflect all these findings. The table or figure should reflect whatever is in the text. The other insignificant data can be included as supplementary data.

Answer: We made a significant changes in the section 2.3